# Regulation of RXR-RAR Heterodimers by RXR- and RAR-Specific Ligands and Their Combinations

**DOI:** 10.3390/cells8111392

**Published:** 2019-11-05

**Authors:** Albane le Maire, Catherine Teyssier, Patrick Balaguer, William Bourguet, Pierre Germain

**Affiliations:** 1Centre de Biochimie Structurale (CBS), CNRS, INSERM, Univ. Montpellier, ICM, 34090 Montpellier, France; lemaire@cbs.cnrs.fr (A.l.M.); catherine.teyssier@inserm.fr (C.T.); bourguet@cbs.cnrs.fr (W.B.); 2Institut de Recherche en Cancérologie de Montpellier (IRCM), INSERM, Univ. Montpellier, ICM, 34090 Montpellier, France; patrick.balaguer@inserm.fr

**Keywords:** RAR, RXR, ligands, retinoids, heterodimers, coactivators, corepressors, transcription

## Abstract

The three subtypes (α, β, and γ) of the retinoic acid receptor (RAR) are ligand-dependent transcription factors that mediate retinoic acid signaling by forming heterodimers with the retinoid X receptor (RXR). Heterodimers are functional units that bind ligands (retinoids), transcriptional co-regulators and DNA, to regulate gene networks controlling cell growth, differentiation, and death. Using biochemical, crystallographic, and cellular approaches, we have set out to explore the spectrum of possibilities to regulate RXR-RAR heterodimer-dependent transcription through various pharmacological classes of RAR- and RXR- specific ligands, alone or in combination. We reveal the molecular details by which these compounds direct specificity and functionality of RXR-RAR heterodimers. Among these ligands, we have reevaluated and improved the molecular and structural definition of compounds CD2665, Ro41-5253, LE135, or LG100754, highlighting novel functional features of these molecules. Our analysis reveals a model of RXR-RAR heterodimer action in which each subunit retains its intrinsic properties in terms of ligand and co-regulator binding. However, their interplay upon the combined action of RAR- and RXR-ligands allows for the fine tuning of heterodimer activity. It also stresses the importance of accurate ligand characterization to use synthetic selective retinoids appropriately and avoid data misinterpretations.

## 1. Introduction

All *trans* retinoic acid (RA) and its synthetic analogs, referred to as retinoids, play crucial roles in a wide variety of biological processes including embryonic morphogenesis and organogenesis, cell proliferation, differentiation and apoptosis, homeostasis, as well as in their disorders [1,2,3,4]. These pleiotropic effects are mediated through retinoic acid receptors (RARs) consisting of three subtypes, α (NR1B1), β (NR1B2), and γ (NR1B3) belonging to the nuclear hormone receptor superfamily [5,6,7,8]. RARs are modular proteins composed of several domains, most notably a DNA-binding domain (DBD) and a C-terminal ligand-binding domain (LBD), acting as ligand-regulated transcription factors by forming heterodimers with the retinoid X receptor (RXR, NR2B) [9,10]. These RXR-RAR heterodimers are the functional entities which bind to specific RA-responsive elements (RAREs) located in target gene promoters and regulate gene expression in a retinoid dependent manner [11,12,13,14]. The basic mechanism for switching on transcription involves a network of interactions with coregulatory protein complexes, the assembly of which is directed by the LBD of receptors. Once bound to RAREs, heterodimers can either repress or activate expression of their target genes by recruiting auxiliary proteins, denoted corepressors (CoRs) and coactivators (CoAs) which modify chromatin and/or interact with the general transcriptional machinery so as to control gene transcription [15,16,17]. RARs in complex with their RXR heterodimeric partner typically recruit CoRs such as SMRT/TRAC [18] or NCoR/RIP13 [19] and repress gene expression in the absence of agonist or in the presence of some antagonists [20,21]. However, when released from CoRs upon agonist binding, they interact with CoAs such as the p160 family members, and activate target gene expression [22,23].

Together with biochemical analyses, structural studies of RXR- and RAR-LBDs and/or their interactions with CoA- and CoR-derived peptides in the presence of various ligands greatly advanced our knowledge of the structural determinants of the interaction between receptors and either type of transcriptional co-regulators both at the level of the individual receptor and in the context of the RXR-RAR heterodimer [24,25]. Following agonist binding, RAR- and RXR-LBDs undergo a conformational change of variable amplitude depending on the receptor type and leading to the stabilization of the activation helix H12 of LBDs in its active position, allowing the recruitment of CoAs [23,26,27]. The p160 proteins share significant sequence homology and are encoded by three distinct genes including the steroid receptor coactivator-1 (SRC1), the transcription intermediary factor 2 (TIF2, also termed SRC2 or GRIP1), and the receptor-associated coactivator 3 (RAC3 also termed SRC3 or AIB1) [28]. These proteins interact directly with the LBD of both RXR and RAR in an agonist-dependent manner by means of short signature sequences located in their nuclear receptor interacting domains (NRIDs). These domains are composed of repeats of LxxLL motifs, in which x is any amino acids and L is leucine, embedded in an α-helical peptide (NR box) [29]. These motifs are both necessary and sufficient for binding to LBDs. NRIDs of the SRC1/TIF2/RAC3 family members contain three LxxLL motifs, conserved in both sequence and spacing [22,30]. Regarding CoR proteins, NCOR, and SMRT share similar domain organizations, interact directly with unliganded-RARs or RAR bound to some antagonists through their NRID and allow the recruitment of various silencing factors such as histone deacetylases (HDACs) or DNA-methyl transferases (DNMTs) [16] that may lead to an inactive condensed chromatin structure preventing transcription [20,31]. The NRIDs of SMRT and NCoR are composed of two or three interaction domains (ID1–ID3) exhibiting sequences (CoRNR1, CoRNR2, and CoRNR3 by analogy with NR boxes of CoAs) similar but not identical to the LxxLL motif of CoAs which were also predicted to adopt an amphipathic helical conformation [32,33]. The structure of the complex formed by the RARα LBD bound to the inverse agonist BMS493 and CoRNR1 of NCoR not only confirmed this prediction, but also revealed that the repressive activity of RARα is conferred by an extended β-strand that forms an antiparallel β-sheet with specific CoRNR1 residues [23]. It was shown that agonist binding induces a β-strand to α-helix transition, which provokes CoR release and CoA recruitment. In addition, the generation of a smaller H3-, H4- and H12-containing surface through the agonist-induced repositioning of RAR LBD helix H12 is also thought to account for the dissociation of CoRs and the recruitment of CoAs via their shorter LxxLL-interaction motifs.

In RAR-containing heterodimers, RXR cannot respond to its ligand unless its heterodimeric partner is bound to an agonist [30,34,35,36,37]. Consequently, RXR-selective agonists cannot trigger RXR-RAR transcriptional response on their own. This phenomenon, referred to as RXR “subordination” or “silencing”, can be best explained by the fact that binding of RAR agonists is both necessary and sufficient to dissociate CoRs from the heterodimer. Hence, RAR apparently “controls” the activity of the heterodimer by (i) silencing RXR activity and (ii) inducing transcription in response to its own ligand. In contrast, when both agonists are used, RXR ligands further stimulate the transcriptional activity of the heterodimer in a synergistic manner. Previous works suggested that synergy originates from the cooperative binding of two LxxLL motifs from one CoA molecule to RXR-RAR heterodimer [22,30].

In terms of ligand binding, RARs bind both all-trans- and 9-cis-RA stereoisomers, whereas RXRs bind only 9-cis-RA [38,39]. Beyond the therapeutic value of retinoid signaling pathways [40,41], there was a great need for selective ligands towards each RAR paralog and RXR with agonist or antagonist activity to allow further dissect the specific function of various receptors using a pharmacological approach in vitro and in vivo, and thus tackle important questions that could not be addressed by gene knockout [42,43,44]. The molecular determinants of the selective interactions between RXR and RAR on one hand, and between all three RAR subtypes on the other hand, are reasonably understood [37,45,46]. It allowed the generation of entirely selective-ligands for all three RAR subtypes and RXR, but also compounds exhibiting complex activities such as retinoids that are antagonists for both RARα and RARγ and agonists for RARβ [21,47]. Moreover, we and others have characterized different classes of retinoids allowing the definition of modulators with various activities based on their ability to differently regulate interactions between receptors and transcriptional co-regulators. These investigations identified molecules with agonistic-, partial agonistic-, antagonistic-, and inverse agonistic-activity [21,22,48]. Strikingly, an RXR antagonist, LG100754 (LG754) [49], was previously described as being able to function as an RXR-RARα activator. It was proposed that LG754 binding to RXR does induce a conformational change in the unliganded-RARα LBD and the recruitment of the CoA SRC-1, raising the question of allosteric communications in RXR-RAR heterodimers [50,51]. Overall, this retinoid collection represents a unique pharmacological toolbox to probe the involvement of each heterodimeric subunit in the specification of RXR-RAR-dependent gene regulation. However, whether the activity of these synthetic retinoids is affected by heterodimerization or not is a crucial question since heterodimers are the functional units that mediate retinoid pathways.

Here, we report on the comparative mechanistic analysis of distinct types of retinoids selective to RARs and/or RXR, and the ability of these compounds, alone or in combination, to differentially modulate the transcriptional activity of RXR-RAR heterodimers. We reveal new functionalities of previously reported antagonists and show that co-regulator binding plays a critical role in the differential response of RXR-RAR to a variety of ligands and that, independent of their binding affinity, these compounds have vastly different properties in controlling the association of co-regulator with heterodimers. We show that heterodimers can be thought of as molecular devices through which precise control of gene transcription can be achieved by using combinatorial sets of ligands, thereby allowing for the initiation of complex gene programs in a cell-specific manner.

## 2. Materials and Methods

### 2.1. Materials and Chemicals

RXR is referred to RXRα hereafter. The pSG5-based Gal-RARα LBD (Gal-RARα), Gal-RXR LBD (Gal-RXR), Gal-TIF2 NRID (Gal-TIF2), Gal-SMRT NRID (Gal-SMRT), Gal-NCoR NRID (Gal-NCoR), RARα LBD-VP16 (RARα-VP16), RXRΔAB-VP16 (RXR-VP16), RXRΔAB, RARαΔAB, RXR, and RARγ expression vectors, and the (17m)5x-βGlob-Luc and the (RARE)3x-tk-Luc reporter genes have been described [52,53,54]. CD3254, UVI3003, LG100754 (LG754), LE135, CD2665, BMS961, BMS614, and BMS493 were from Tocris. BMS948 was provided by Bristol-Myers Squibb (New York, NY, USA) and AGN192870 (AGN870) by Galderma (Lausanne, Switzerland). UVI3002 was a gift of Angel de Lera (University of Vigo, Vigo, Spain). Am580 and TTNPB were from Sigma France (Saint Quentin Fallavier, France).

### 2.2. Transient Transfections

Transient transfections and two-hybrid assays were performed as described. Briefly, COS cells were cultured in Dulbecco’s Modified Eagle Medium (DMEM) with Glutamax and 10% (*v*/*v*) fetal calf serum (FCS) and transfected using either JetPei transfectant (Ozyme, Saint-Cyr-l’Ecole, France) or the standard calcium phosphate precipitation technique. After 24 h, the medium was changed to a medium containing the indicated ligands or vehicle. Cells were lysed and assayed for reporter expression 48 h after transfection. The luciferase assay system was used according to the manufacturer’s instruction (Promega, Madison, WI, USA). In each case, results were normalized to co-expressed β–galactosidase.

### 2.3. MRLN Cell Line Generation

The stably transfected RARE reporter cell line MRLN was generated as previously described [55]. Briefly, MCF-7 cells at 30–40% confluence, grown in DMEM complemented with 10% FCS, were transfected by the standard calcium phosphate precipitation technique with the (RARE)3x-tk-Luc-Neomycin reporter gene. At 48 h afterwards, geneticine (G418) was added to a final concentration of 1 mg/mL. G418-resistant RA-responsive clones were purified by limiting dilution in the presence of G418 selection. Upon confluence, cells were transferred into 24-well plates and inducibility was re-tested. Only highly inducible cell clones were expanded, regularly checked for inducibility and aliquots frozen at different passages.

### 2.4. Cell Proliferation Analysis

MCF-7 cell proliferation was evaluated by growth curves. Cells were cultured in DMEM with Glutamax and 10% (*v*/*v*) FCS and seeded in 6-well tissue-culture plates at a concentration of 10^4^ cells/well with 2 mL of specific medium. Cells were allowed to attach for 24 h and then incubated in medium with the appropriate concentration of ligands. Control cells were treated with ethanol alone (final concentration, 0.1%). The medium was renewed every 2 days. Cell numbers were determined by counting the cells at the start of the experiment and on the sixth day. Each experiment was performed in triplicate.

### 2.5. Cell Differentiation Analysis

F9 cells were maintained in DMEM with Glutamax and 10% (*v*/*v*) FCS. For differentiation studies, F9 cells were seeded in 6-well tissue-culture plates coated with gelatin (0.1%) at a concentration of 10^4^ cells/well with 2 mL of specific medium and were treated with ligands for 72 h with a change of medium after 48 h. Control cells were treated with ethanol alone (final concentration, 0.1%). Morphological differentiation was monitored and expressed as percentage of the cell population exhibiting a differentiated phenotype.

### 2.6. Electrophoretic Mobility Shift Assay

The TnT® Quick Coupled Transcription/Translation System (Promega, Madison WI, USA) was used to produce RXRΔAB and RARαΔAB proteins in vitro. TIF2 (TIF2.42 residues 624–828) and SMRT (SMRTct residues 982 to end) were purified as previously described. The RXRΔAB-RARαΔAB heterodimers were incubated with saturating amounts of the respective ligands for 30 min at 4 °C. Purified co-regulators (TIF2 and SMRT) were added and incubated for a further 15 min at 4 °C. The heterodimer–co-regulator complexes were further incubated for 15 min with 25,000 c.p.m. pre-annealed RARE that was labelled with ^32^P (5′-TCGAGGGTAGGGGTCACCGAAAGGTCACTCG-3′ direct repeat underlined) oligonucleotide at 4 °C. The total volume of the entire reaction mix was 22 mL. The protein-DNA complexes were resolved on non-denaturing 6% polyacrylamide gels in 0.5× TBE buffer for 3 h (pre-run overnight at 200V, 4 °C). The gels were dried and subjected to autoradiography.

### 2.7. Limited Proteolytic Digestion

In vitro-made ^35^S-labelled human RARs (TNT kit, Promega, Madison, WI, USA) were used for limited proteolysis as described previously [53,56]. Briefly, receptors proteins were incubating on ice for 60 min in the absence of in the presence of ligands at different concentrations, and then digested at 25 °C for 10 min with 100 µg/mL (RARα and RARβ) or 50 µg/mL (RARγ) of trypsin.

### 2.8. Crystallization of the RXRα LBD-LG754 and RARβ LBD-LG754 Complexes

Expression and purification of the human RXRα and RARβ LBDs have been described previously [37,54]. Fractions containing the purified receptor were pooled, mixed with a threefold molar excess of LG754 and a threefold molar excess of the TIF2 NR2 (for RXRα complex) or SRC1 NR2 (for RARβ complex) coactivator peptide and concentrated to 10 mg/mL. Crystals were obtained by vapor diffusion at 293 K. The well buffer contained 17.5% PEG 3350, 0.7 M ammonium acetate and 30 mM Na acetate pH4.6 or 200 mM Tri Sodium Citrate pH 5.5, 25% PEG 4000 for RXRα or RARβ complexes, respectively. Crystals grew in a few days and were of space group P43212 or P212121 for RXRα or RARβ complexes, respectively. For each complex, a single crystal was mounted from the mother liquor onto a cryoloop, soaked in the reservoir solution containing an additional 20% glycerol and frozen in liquid nitrogen.

Crystallographic data collection, processing and structure refinement. Diffraction data were collected at the ID14-1 and ID23-1 beamlines of the European Synchrotron Radiation facility (ESRF, Grenoble, France) at 1.9 and 2.3 Å resolution, for RXRα and RARβ complexes, respectively. Diffraction data were processed using MOSFLM [57] and scaled with SCALA from the CCP4 program suite [58]. The structure was solved by using the previously reported structures 3E94 [59] or 4JYI [37], for RXRα or RARβ complexes, respectively, from which the ligand was omitted. Initial *F*_o_–*F*_c_ difference maps showed a strong signal for the ligand, which could be fitted accurately into the electron density. The structure was modelled with COOT [60] and refined with REFMAC [58] using rigid body refinement, restrained refinement, and individual B-factor refinements.

## 3. Results

### 3.1. CD2665 Is an RARα Antagonist

In the past, major research efforts have been directed to the identification of potent synthetic retinoids leading to the generation of a panel of modulators exhibiting or not selectivity towards RAR subtypes, and with activities ranging from agonist to inverse agonist [43] [21,44]. Among these modulators, CD2665 has been reported as a specific RARγ/β antagonist (Figure 1).

However, the Kd values reported were > 1000 nM, 400 nM, and 81 nM, for RARα, RARβ, and RARγ [61], respectively, showing that CD2665 has a preference for both RARγ and RARβ but is also able to bind to RARα when used at micromolar concentration. Transient transactivation assays were conducted in order to assess the ability of CD2665 to affect TTNPB-induced RARα activity. In this assay, cells expressed a fusion protein comprising the LBD of RARα and the DBD of the yeast GAL4 transcription factor. Note that this reporter system is insensitive to endogenous receptors which cannot recognize the GAL4-binding site. In these experiments, the full RAR agonist TTNPB at 3 nM was challenged with increasing amounts of CD2665 for which the antagonistic activity was compared to that of the selective RARα antagonist Ro41-5253 [62] (Figure 2A). Competition curves showed that CD2665 exhibits a clear dose–response inhibitory effect on TTNPB-induced RARα activity, but with an IC50 higher than that of Ro41-5253, in line with their respective affinity reported for RARα. These results show that, in addition to its antagonistic activity toward RARγ and β, CD2665 acts as an RARα antagonist.

### 3.2. Co-Regulator Interactions Define Different Classes of RARα Antagonists

In previous reports, two-hybrid systems involving both CoA and CoR allowed the definition of the molecular basis of the agonistic (TTNPB), partial agonistic (AGN870), neutral antagonistic (BMS614), and inverse agonistic (BMS493) activity of ligands [9,21,22]. In this context, both CD2665 and Ro41-5253 need to be better characterized in terms of ability to modulate transcriptional co-regulator interaction with RARα. To this end, we used a standard mammalian two-hybrid assay comprised of the TIF2 NRID fused to the Gal DNA binding domain of the yeast *Saccharomyces cerevisiae* (Gal-TIF2) as bait, the RARα LBD fused to the *Herpes simplex* VP16 acidic transcription activation domain as prey (RARα-VP16), and the corresponding reporter system (17m)x5-βGlob-Luc. In this assay, Gal-TIF2 specifically binds to the ‘‘17m’’ DNA recognition site through the Gal DBD and can interact with the RARα LBD in the presence of RARα agonists. As the VP16 domain confers constitutive transcription activation if it is brought close to a promoter, a specific gene induction is seen only if the two proteins bind to each other. As expected, addition of the RAR-specific agonist TTNPB promoted an interaction between RARα and TIF2, whereas all the antagonists did not support efficient CoA interaction (Figure 2B). However, a weak activation could be observed in the presence of CD2665 or AGN870. Strikingly, in another assay where the CoA RAC3 was overexpressed, the agonistic activity of TTNPB, and the weak transcriptional activity of both CD2665 and AGN870 could be largely enhanced, suggesting that the interaction surface between CoA and RARα was not fully arrested in the presence of CD2665 and AGN870 (Figure 2C). In contrast, both BMS614 and BMS493 weakly affected the ability of RARα to associate with CoAs in comparison to the basal level. Interestingly, the complex RARα-Ro41-5253 was insensitive to changes in RAC3 expression levels, and even it totally impaired CoA interaction when compared to activity measured in the absence of any retinoid. This observation suggested that Ro-415253 must induce a conformation of the RARα-LBD totally incompetent to interact with CoAs.

In addition, two-hybrid analyses with the CoRs SMRT and NCoR revealed that retinoids displayed very divergent ability to modulate the association of CoRs with RARα, and confirmed the definition of TTNPB, BMS493, and BMS614, as agonist, inverse agonist, and neutral antagonist, respectively (Figure 2D). While TTNPB and BMS493 were able to decrease and enforce CoR association, respectively, BMS614 promoted a partial reduction of SMRT interaction and no significant effect was seen on NCoR binding with this molecule. SMRT and NCoR binding was rather decreased in the presence of all the antagonists CD2665, AGN870, and Ro42-5253. Relative to TTNPB, the effect of CD2665 on the interaction of CoRs with RARα was more pronounced for SMRT than for NCoR. Overall, our data confirmed that retinoids can be classified according to their ability to modulate co-regulator association with RARα and revealed that CD2665 is able to act as an RARα antagonist being capable of dissociating the CoR without generating an efficient CoA-binding surface.

### 3.3. Synergistic Activation by RARα Antagonists and a Rexinoid Agonist to Activate RXR-RARα Heterodimer

We then investigated whether the functional features of the retinoids characterized above in a monomeric context were affected by heterodimerization, thereby leading to new ligand functionalities. To address these questions, transient co-transfection experiments in COS cells were performed with a luciferase reporter gene driven by a RARE sequence in front of the thymidine kinase promoter (RARE)3x-tk-Luc and expression vectors for both RXR and RARα deleted of their AB domains (referred to as RXRΔAB and RARαΔAB) (Figure 3A). These transfected COS cells were exposed to various RAR modulators alone or combined with the RXR agonist CD3254. Whereas TTNPB activated transcription through RXRΔAB-RARαΔAB heterodimer, the RXR agonist CD3254 alone was inactive, in accordance with the subordination model. Together, both compounds transcriptionally cooperated, as the level of gene expression achieved was higher than that induced by TTNPB alone. None of the above characterized antagonist was able to exert a significant effect when used alone. In a striking contrast, co-treatment of the various RARα ligands with CD3254 yielded different transcriptional responses. Notably, co-treatment with BMS493, BMS614, and Ro41-5253 remained ineffective to activate transcription. On the contrary, both CD2665 and AGN870 in the presence of CD3254 promoted a level of expression of the reporter gene comparable to that produced by TTNPB alone. A similar pattern of activation with these ligand combinations was obtained in transient transfection assays using full-length receptors (data not shown). These results demonstrate that RARα antagonists can be differentiated on the basis of their ability to transcriptionally synergize with an RXR agonist and that the LBDs of RXR and RARα are required and sufficient to reveal the transcriptional synergy between the two partners.

Subsequently, the MCF-7 breast cancer cell line was chosen to investigate whether a response pattern observed for heterodimer lacking AB domains in transient transfection experiments could be reproduced with an endogenous level of RXR-RARα heterodimers. Indeed, several laboratories have concluded that, despite the expression of RARγ, RARα is the receptor that selectively relays retinoid-mediated signaling in MCF-7 cells [63]. In particular, treatment of MCF-7 cells with retinoids capable of discriminating among RAR subtypes have confirmed that RARα is the mediator of the anti-proliferative effect of RA in MCF-7 cells [64]. To evaluate the possible RXR-RARα-dependent transactivation in these cells, we established a MCF-7-derived cell line, denominated MRLN, which stably contains the luciferase reporter system (RARE)3x-tk-Luc [55,65]. Importantly, MRLN assays gave similar results to those obtained with the transient transfection using RXRΔAB-RARαΔAB (Figure 3B). Furthermore, comparison of the transcriptional activity of RAR ligands in combination or not with CD3254 with their growth-inhibitory activities after six days of culture revealed a strict correlation between the two assays (Figure 3C). Clearly, the retinoids BMS493 and BMS614 that did not transcriptionally synergize with CD3254 were not inhibitors of the MCF-7 cell growth. However, Ro41-5253 that did not activate RXR-RARα transcriptional function slightly induced MCF-7 cell growth inhibition [63]. This activity is likely independent from the RARα pathway but can rather be attributed to the ability of this compound to bind and activate another nuclear receptor, PPARγ [66]. Anyhow, the combination of Ro41-5253 and CD3254 was no more effective on MCF-7 cell proliferation, which is in line with the inability of both molecules to transcriptionally cooperate. Clear synergistic effects between CD2665 or AGN870 and CD3254 on MCF-7 cell proliferation were observed as these combinations displayed degrees of growth inhibition similar to that observed with TTNPB alone. Nevertheless, these co-administrations did not reach the inhibition produced by TTNPB-CD3254 co-treatment, in keeping with the level of transcription measured with the MRLN reporter cell line. Importantly, this cellular assay demonstrated that the synergy occurring between RARα antagonists and RXR agonist can happen in a cellular environment and that it was efficient enough to initiate the entire RXR-RARα-mediated genetic program leading to a reduction of the cellular growth, which paralleled the luciferase profile expression in MRLN cells.

To decipher the mechanism by which some RARα antagonists enable RXR-RARα heterodimer activation by rexinoids, we examined whether these compounds allow pharmacological modulation of heterodimer-co-regulator complexes formation by electrophoretic mobility shift assays (EMSAs) (Figure 3D). Incubation of RXRΔAB-RARαΔAB heterodimer bound to the RARE DNA sequence with a mix of SMRT and TIF2 produced a heterodimer-SMRT complex detectable as a super-shift. SMRT, which dissociated and was replaced by TIF2 as expected in the presence of the RAR agonist TTNPB, remained bound to a heterodimer when exposed to the RXR agonist CD3254 alone despite the presence of TIF2, in accordance with the subordination principle. Remarkably, CD2665 alone was capable of dissociating SMRT without generating a surface efficient enough to enable a stable CoA association, so essentially generating a naked heterodimer. Strikingly, CD2665-CD3254 combination promoted heterodimer-TIF2 interaction. This suggested that, in this case, the active state of heterodimer is achieved by the ability of CD2665 to release SMRT from RARα and of CD3254 to promote association of TIF2 with RXR. Hence, this combination permits the transition from the repressive to the active state which parallels its ability to transactivate through a classical heterodimer-RARE-mediated pathway (Figure 3A,B). Interestingly, despite decreasing the amount of bound SMRT, Ro41-5253 completely blocked TIF2 interaction either by itself or in association with CD3254, in keeping with its inability to transcriptionally cooperate with an RXR agonist and its capability of blocking CoA binding. Finally, AGN870 when used alone generated equilibrium between the three possible complexes, namely naked heterodimers or heterodimers associated with either TIF2 or SMRT. CD3254-AGN870 co-incubation produced a dominant band corresponding to the heterodimer-TIF2 complex, suggesting that CD3254 addition displaced the equilibrium between the various complexes towards this configuration. It is noteworthy that these interaction patterns produced by RAR ligands recapitulate those defined above for the RARα monomer in two-hybrid experiments, indicating the preservation of ligand properties in the context of the heterodimer (Figure 2B–D). Together, these data suggest a model according to which the RXR agonist can induce CoA recruitment by the heterodimer through RXR if the heterodimerization partner RAR is bound to a compound that destabilizes CoR interaction and does not completely block that of CoA. Then RXR-RARα heterodimers integrate the signaling capacity of the two subunit LBDs into a combined transcriptional response that allows for the fine-tuning of gene expression.

### 3.4. Synergistic Activation by RARγ Retinoids and a Rexinoid Agonist to Activate RXR-RARγ Heterodimer

Because our above set of analyses demonstrated that synergistic activation was detected upon CD2665-CD3254 co-treatment and that RAR ligands kept their properties towards RARα in the context of the heterodimer, we wanted to know whether the same was true with the RARγ subtype. Transient co-transfection experiments in COS cells performed with the RAR reporter gene (RARE)3x-tk-Luc and both RXR and RARγ expression vectors revealed that co-treatment with CD2665 and CD3254 did not show any significant effect on the expression of RXR-RARγ-mediated gene expression (Figure 4A). This is in marked contrast with the results obtained with the RXR-RARα heterodimer. The same observation was made for the other antagonists, including AGN870 and BMS493, while cooperativity between RXR and RARγ occurred when using CD3254 with either TTNPB or the RARγ partial agonist LE135. Importantly, these transcriptional activities paralleled the ability of these compounds to modulate the interaction of CoRs with RARγ as demonstrated by two-hybrid experiments, which revealed that both CD2665 and AGN870 were unable to dissociate SMRT from RARγ while TTNPB and LE135 provoked a major or partial release, respectively (Figure 4B).

In order to determine if the transactivation data observed in transfection experiments held true for endogenous RXR-RARγ heterodimers in a more physiological context, we repeated the experiments in the F9 murine embryonal carcinoma cell line [11]. This cell line provides a useful model for analysis of RXR-RARγ functions at the cellular and molecular level, since it was proven that RXR-RARγ heterodimers are the functional units mediating the effects of retinoids in triggering target gene activation and differentiation into primitive endoderm in F9 cells [67]. We then investigated whether ligands were able to promote F9 cell morphological differentiation and found that differentiation occurred with ligands or combination of ligands which activated RXR-RARγ heterodimer (Table 1). In this respect, only exposure to the RARγ agonists (TTNPB and BMS961) induced morphological changes after three days of treatment, whereas CD3254, CD2665, and AGN870 were inefficient in triggering differentiation. Notably, under CD2665-CD3254 or AGN870-CD3254 co-treatment F9 cells retained their undifferentiated morphology, confirming that this combination was inefficient in activating RXR-RARγ, whereas addition of CD3254 was able to potentiate the LE135-induced differentiation in accordance with the above transactivation data (Figure 4A). Taken together, our results support the idea that, in the same way as for RXR-RARα heterodimer, RAR ligands retain their functional features towards RARγ and dictate the RXR response to its ligand in the context of the RXR-RARγ heterodimer. On the other hand, in the presence of an RXR agonist, CD2665 was able to mediate transactivation through RXR-RARα heterodimers, whereas it blocked the activity of heterodimers containing RARγ. This demonstrates that it is possible to generate retinoids modulating heterodimer activity in a RAR subtype and co-regulator-selective manner and that the acquired pattern of coregulatory interaction apparently accounts for their activity.

### 3.5. Contribution of Different Classes of Rexinoids in the RXR-RAR Heterodimer Activity

Having shown that RAR ligands retained their binding ability and modulation properties of co-regulator interaction with the RAR receptor in the context of RXR-RAR heterodimers, we then investigated if the same was true for RXR and its ligands. To this end, we took advantage of a previously developed series of RXR modulators derived from the full RXR agonist CD3254 [54,68]. In the current study, we used the mixed agonist/antagonist UVI3002 and the full antagonist UVI3003. On the other hand, the modulator LG754 has been reported as an RXR antagonist capable of activating the heterodimer RXR-RARα. In order to better characterize the functional profile of this ligand, we performed transactivation experiments with COS cells transiently co-transfected with a vector expressing the chimeric protein Gal-RXR LBD and the corresponding reporter system (17m)x5-G-Luc. It should be noted that similar results were obtained with alternative models using full-length RXR (data not shown). Dose-dependent competition curves confirmed that UVI3002, UVI3003, and LG754 efficiently antagonized RXR-CD3254-induced transcription (Figure 5A). The IC50 differences presumably reflected a higher affinity of LG754 for RXR. Subsequently, two-hybrid experiments similar to those performed with RAR were conducted to determine the ability of these ligands to modulate the interaction of co-regulators with RXR. Figure 5B shows that, in the absence of any added ligand, RXR interacted very weakly with both SMRT and NCoR and that none of the compounds tested exerted a significant effect on these interactions. In a second step, similar assays were performed with the previously published chimeric peptide derived from NCoR, called B [32], and combining the N terminus of CoRNR1 with the C terminus of CoRNR2, which was proven to possess greater ability to interact with nuclear receptors than wild-type NCoR. While CD3254 had no effect on B peptide interaction with RXR, UVI3003 promoted a strong increase in B peptide association, UVI3002, and LG754 having an intermediate effect. This result suggested a correlation between the length of the UVI3003 bulky chain (Figure 1) and the ability of this compound to help B peptide recruitment by unlocking an interaction surface accessible for the B peptide on RXR. In conclusion, all tested RXR ligands did not promote a gain in CoR binding, meaning that these compounds do not act as RXR inverse agonist and do not generate a binding surface for SMRT and NCoR. Two-hybrid experiments exploring TIF2 interaction with RXR showed that the full RXR agonist CD3254 induced robust TIF2 association, whereas both LG754 and UVI3002 did so only weakly and UVI3003 was inactive (Figure 5C). However, the weak agonist activity of both LG754 and UVI3002 could be largely enhanced when the CoA RAC3 was overexpressed (Figure 5C). Together, these data demonstrate that all these rexinoids, whether agonists, partial agonists, or full antagonists, have no influence on the interaction of RXR with CoRs, and that their activating capacity relies only on their ability to induce the recruitment of CoAs. Note that LG754 and UVI3002, which harbor the same bulky chain, were poorly distinguishable from each other, LG754 being slightly more antagonistic than UVI3002.

Our above results showed that an RXR agonist can participate in the activation of RXR-RAR heterodimers when combined with an RAR ligand capable of decreasing the interaction of CoR. We then evaluated the ability of other classes of rexinoids to modulate the activity of the RXR-RAR heterodimer alone or in combination with either TTNPB or AGN870 by using the MRLN cell model (Figure 5D). When used alone, CD3254 and the two UVIs did not significantly activate RXR-RARα. Importantly, and in accordance with previous reports [50,51], LG754 acted as an activator of the heterodimer by reaching a transactivation level of about 70% of that induced by the full RAR agonist TTNPB. Added at 1 µM to fully saturate RARα LBD and in combination with TTNPB, both UVI3002 and LG754 caused a slight over-activation compared to TTNPB alone, but less than the increase observed with CD3254, while the antagonist UVI3003 had no effect. Similar cooperativities were observed with AGN870 at a lower extent due to the antagonistic nature of this RAR compound. Together, these data show that the two heterodimer subunits retain their ligand and co-regulators binding properties, and that rexinoids can contribute to heterodimer activity in combination with an appropriate RAR ligand. Binding of a RAR ligand capable of dissociating CoRs is a prerequisite for the activation of the heterodimer.

### 3.6. LG754 Is a Partial Agonist Relative to RARs

Besides the activation induced by LG754 in the MRLN system, we observed that co-incubation with the RAR antagonist AGN870 led to a decrease in the transactivation caused by LG754 alone (Figure 5D). Although it has been previously proposed that binding of LG754 to RXR allosterically activates RAR (a phenomenon referred to as “the phantom effect”) [50,51], the aforementioned observation suggests that LG754 could also bind to RAR, and be competed off by AGN870. Consequently, we carried out experiments to investigate whether this modulator is capable of interacting with the three RAR subtypes. Limited proteolysis was conducted because a conformational change in the RAR LBD occurs after ligand binding causing greater resistance to trypsin digestion. As a result, proteolysis assays can be used to monitor ligand binding and evaluate binding affinity as the proteolytic resistance of protected fragments is dependent on ligand concentration [53]. Limited trypsin digestion assays of in vitro translated RARα, β, and γ in the presence of either TTNPB, LG754 or CD3254 were performed. Note that TTNPB was used in a range of concentrations from 10 nM to 10 µM, instead of 1 nM to 100 nM for both LG754 and CD3254 (Figure 6A). Specifically, under our experimental conditions, the first effects were observed at TTNPB concentrations around 10 nM, in keeping with the high affinity of this ligand for RARs, and the highest degree of resistance was achieved at 100 nM. In contrast, no significant CD3254-dependent protection of RARs was observed, protection starting only at the very high concentration of 100 µM, confirming the inability of this molecule to significantly bind to RARs. Importantly, RARs showed LG754-dependent sensitivity from around 100nM, indicating a LG754-induced conformational change of RAR LBDs as a result of LG754 binding, but also a difference of relative affinity of around two orders between LG754 and TTNPB. Interestingly, LG754-RAR LBD complexes were more sensitive to proteolytic digestion than TTNPB suggesting that these two compounds differently stabilized RAR LBDs. Taken together, partial proteolytic digestion with trypsin supported the idea that LG754 binds to all three RAR paralogs but with a lower binding affinity than TTNPB.

In order to know if the demonstrated LG754 binding to RARs translated into a transcriptional effect, we performed reporter assays using the Gal-RAR system (Figure 6B). Retinoid agonists discriminating between RAR subtypes (Am580, BMS948, and BMS961 for RARα, β, and γ, respectively) were used as control of the response selectivity [37]. Strikingly, these analyses clearly showed that 1 µM LG754 was efficient for transactivation in the three RAR paralog conditions. Both CD3254 and UVI3002 were inactive, in line with their inability to bind RARs. While some compounds exhibited complex activity such as AGN870 which was RARα/γ antagonist and partial agonist for RARβ or LE135 displaying a partial agonist for RARγ without activating RARα and RARβ, LG754 displayed a partial agonistic activity for all three RARs. This latter observation may be related to the lower ability of LG754 compared to TTNPB in triggering F9 morphological differentiation (Table 1). It is noteworthy that exposure of F9 cells to CD3254-LG754 co-treatment promoted differentiation more efficiently than LG754 alone and as efficiently as TTNPB, indicating a cooperativity between CD3254 and LG754 for RXR-RARγ heterodimer activation.

To further decipher the activity of LG754 in RAR, interaction of co-regulators with the LG754-RARα LBD complex was assessed using two-hybrid assays (Figure 6C,D). As expected, the pure rexinoids, CD3254, and UVI3002, failed to promote co-activator interaction with RAR. In striking contrast, LG754 induced the recruitment of TIF2 almost as efficiently as TTNPB, while it provoked a partial dissociation of the two CoRs from RARα. Moreover, LG754 was able to counteract CoR release or increase induced by TTNPB or BMS493, respectively, in a dose–response manner. This observation supports the notion that LG754 can efficiently compete with RARα ligands for LBD binding (Figure 6E,F).

Overall, demonstration has been done that, in addition to its RXR activity, LG754 binds to all three RAR paralogs and acts as an RAR partial agonist. Its partial activity results from its ability to shift the balance of transcriptional co-regulator interactions to RARs in favor of CoAs versus CoRs, which results in an activation of RXR-RAR heterodimers. Hence, LG754 must be qualified as a potent and weak partial agonist for RAR and RXR, respectively.

### 3.7. Structural Analysis

To provide structural evidence of LG754 activity toward RXR and RAR, we crystallized RXRα or RARβ LBDs in the presence of LG754 and of a coactivator-derived peptide. The crystal structures of the ternary complexes with RXRα and RARβ were solved at 1.9 and 2.3 Å resolution, respectively (Appendix A).

Both LBDs adopt the canonical agonist-bound (active) conformation with the C-terminal activation helix H12 sealing the ligand binding pocket (LBP) and the coactivator peptide bound to the surface generated by helices H3, H4 and, H12 (Figure 7A,B). LG754 ligands could be precisely placed in their respective electron density in the two LBDs as shown by the experimental omit electron density maps (Appendix A), confirming that LG754 can bind to both RXR and RAR. The crystal of LG754-bound RARβ LBD contains two complexes in the asymmetric unit and, while the two RARβ molecules superpose very well (rsmd = 0.410 on 201 Cα) (Figure 7C), LG754 adopts a different conformation in each of them (Appendix A). In both cases, LG754 is stabilized in the RARβ LBP through extensive van der Waals contacts and a network of ionic and hydrogen bonds between the carboxylate moiety of LG754, R269 in H5 and S280 in the β-turn (Figure 8A,B). However, the propoxy group of the ligand points either toward helices H6 and H7 (Figure 8A) or toward helix H12 (Figure 8B). Both ligand positions are compatible with an agonist conformation of RARβ LBD even though the orientation where the bulky group points toward H12 might destabilize slightly the active position of the activation helix, thus explaining why LG754 is not a full RAR agonist. Comparison of the LG754-bound RXRα LBD structure with that of the previously reported RXRα LBD bound to the full agonist CD3254 [68] reveals some residue reorientations, the most significant one affecting L436 in helix H11 (Figure 8C). In the presence of CD3254, this leucine has a pivotal role in inducing the sharp turn on the LBP volume to accommodate the twisted ligand and in stabilizing the active conformation of H12. In contrast, our structure shows that L436 undergoes a significant conformational change and rotates toward H12 to accommodate the propoxy group of LG754. This conformer generates a steric clash with L455 from helix H12 accounting for the destabilization of the active conformation of H12 in solution. Such a mechanism has already been observed for the partial agonist UVI3002 [54], which displays a similar aliphatic extension projecting toward H12 (Figure 8D). In both cases, the bulky side chains act as antagonistic extensions, lowering the interaction strength between H12 and the LBD surface and accounting for the very weak agonistic profile of these ligands. This is in line with a previous crystal structure of RXRα in complex with LG754 [69] showing a non-active RXRα LBD conformation with the C-terminal H12 flipping out to the solvent (Figure 7D). As already observed with 9CRA [70,71] (Figure 3A), LG754 is able to adopt an extended conformation in RARβ (Figure 9B,C) and a bent conformation in RXRα (Figure 9B,D). As a consequence, the tetrahydronaphtalene group of LG754 points toward helix H12 in RARβ, whereas, in RXRα, a rotation by about 90° around the C9-C10 orients the ring away from helix H12 (Figure 8).

In conclusion, our structural analysis shows that, as with 9CRA, LG754 can bind to both retinoid receptors by adopting a linear I shape conformation in RAR and a shorter L shape conformation in RXR. It also supports our functional data defining LG754 as very weak RXR partial agonist and a rather potent RAR agonist.

## 4. Discussion

Comparative mechanistic analysis of distinct types of both retinoid and rexinoid modulators allowed us for a better definition of the relative contributions of each subunit in the transcriptional state of RXR-RAR heterodimers. Altogether, our results support a model in which each receptor of the heterodimer retains its own properties in terms of ligand and transcriptional co-regulator binding. As a result, RXR-RAR heterodimers integrate the signaling capacity of the two LBDs into a combined transcriptional response. This substantiates that, in contrast to homodimerization, heterodimerization allows for the fine-tuning of nuclear receptor action by using combinatorial sets of ligands. In this context, RXR plays a unique central role within the nuclear receptor family because it is required as an obligate heterodimeric partner for many other receptors implicated in a multitude of pathways [72,73,74,75,76]. However, in RXR-RAR heterodimers, our data confirm that RXR cannot respond to its ligand unless RAR is liganded [36]. Consequently, RXR-selective ligands on their own cannot trigger RXR-RAR heterodimer-mediated retinoid-induced events in various cell systems in line with our MCF-7 and F9 cell based assays [34,35]. This RXR subordination may be of utmost biological importance because it avoids confusion between signaling pathways. The molecular mechanism responsible for this subordination has been a very controversial issue. However, several studies showed that transcriptional co-regulator interactions could account for the inability of RXR to respond to its agonist, leading to the current prevalent point of view supported by our results [22,30,77]. In this model, RXR can bind its own ligand, even in the absence of RAR ligand, but it fails to dissociate the CoR-heterodimer complex. In contrast, binding of an RAR agonist is both necessary and sufficient to dismiss the CoR complex and recruit CoAs, facilitating alterations to local chromatin architecture and subsequent recruitment of the general transcription machinery [22,77,78].

Previous studies allowed us to demonstrate that, in the context of isolated RXR-RAR heterodimer, liganded RXR can actively participate to CoA association by directly recruiting it in the absence of an RAR ligand [22]. However, in the presence of both co-regulator types, the heterodimer remains associated with CoR in keeping with our EMSA data. As the binding of CoRs and CoAs are mutually exclusive, a rexinoid agonist-RXR-RAR complex cannot bind a CoA in the usual cellular environment. This phenomenon presumably originates from a competition between both co-regulators for interaction with heterodimers. In this respect, affinities of CoR for unliganded-RAR or -heterodimer were shown to be higher than that of CoA for liganded RXR or liganded-RXR-unliganded-RAR heterodimer [23,71], thus accounting for RXR subordination. Consequently, a critical point relies on the relative affinities of co-regulators for RXR and RAR which depend on the ligand binding status of each monomer and can reasonably be related to the activation state of the heterodimer. Furthermore, the system is complicated by the presence of two or three NR boxes into NRIDs of co-regulators [28,29,32], making it such that an interplay between heterodimeric subunits may exist. This is illustrated by the fact that, in the presence of both RAR and RXR ligands, synergy requires two LxxLL motif containing NR boxes of one CoA molecule and likely originates from the simultaneous establishment of two heterodimer-CoA interfaces [22,30]. On the other hand, it was recently reported that CoR forms through its two NR boxes a transient multi-site complex with heterodimer and that, even if RAR is the prime contact point of the heterodimer with the CoR, RXR plays a minimal role in its recruitment as well [79]. This latter observation makes that the cooperativity between RXR and RAR is more dramatic for CoA binding than that for CoR. All together, these analyses lead to the conclusion that distinct receptors appear to exhibit different co-regulator stoichiometries and interaction patterns that can be modulated by the use of a particular ligand. As a result, the above considerations together with data reported in the present report highlight a model in which heterodimers are in an equilibrium between repressing and activating states which is modulated by each partner and their respective ligands. The equilibrium is determined by the relative strengths of interacting surfaces of each LBD with co-regulators. For instance, the association of the RAR antagonist CD2665 with the RXR agonist CD3254 results in an activated RXR-RARα heterodimer. By its ability to release CoR from RAR and recruit CoA through RXR, this combination promotes a shift of the balance in favor of CoA and the transition from the repressive to the active states which translates into transcription activation. On the other hand, our results show that the molecular basis of the modulation of RXR-RARα activation applies to heterodimer containing RARγ, namely that both heterodimeric partners keep their own properties related to ligands and co-regulators, and also in term of ability to transcriptionally synergize under exposure to co-treatment with agonists for each partner. However, dramatic differences between these two types of heterodimer can be observed as the inability of CD2665 to synergize with an agonist rexinoid in RXR-RARγ. This failure of synergy is due to the inability of this antagonist to release CoR from RARγ while the partial RARγ agonist LE135 reduces CoR association and allows for a transcriptional synergy. This confirms the necessity of CoR release from RARγ to permit RXR activation and synergy in the context of the RXR-RARγ heterodimer. As a result, it can be stressed that, besides classical RAR-subtype specific ligands, the CD2665-CD3254 combination offers an alternative way to activate RARα in a selective fashion. In addition, as the transcriptional co-regulators are platforms for complex epigenetic activities, such ligands represent valuable tools to regulate RXR-RAR-mediated gene programs in very different directions. In this way, the identity and sequential establishment of various co-regulator complexes may contribute to cell- and receptor-specific gene programming.

Because RAR governs the CoR binding status of the heterodimer and, consequently, the ability of RXR to respond to its own ligand, the degree of CoR interaction is a crucial determinant on which pharmacological agents can act for modulating heterodimer activity. This postulation is supported by our data showing that the pharmacological profiles of retinoids are primarily determined by their impact on CoR interactions. Among these various types of modulators, RAR inverse agonists are defined by their ability to reinforce CoR interaction. AGN193109 was the first reported inverse agonist for RARs [48,80], but it exhibits a weaker inverse agonistic activity than BMS493 [21]. To our knowledge, the latter is the most powerful inverse agonist for all three RAR subtypes. The resolution of the crystallographic structure of the complex formed by the RARα LBD bound to BMS493 and CoRNR1 peptide of NCoR showed that the specificity of the interaction between RAR and CoR is conferred by an extended β-strand in RAR LBD forming an antiparallel β-sheet with CoR residues [23]. It also provided a structural basis for the increase of CoR affinity in the presence of an inverse agonist. Accordingly, RAR inverse agonists prevent agonist-bound RXR from interacting with CoA. In a striking contrast, CD2665 decreases the CoR interaction with RARα and allows liganded-RXR to associate with CoA which produces an active heterodimer. Structurally, bulky groups conferring the antagonistic nature of CD2665 and BMS493 are very different (Figure 1). One can reason that the particular structure of CD2665 impairs the formation of the extended β-strand required for CoR association without generating an optimal surface for CoA interaction. Other retinoids used in this study exhibit an intermediate functional profile such as AGN870. Interestingly, both EMSA and two-hybrid assays with RARα show that, when compared to CD2665, AGN870 is less efficient to release CoR but a little more effective for CoA recruitment. It is likely that the resulting co-regulator equilibriums promoted by the two molecules are similar as they can synergize in a similar manner with an RXR agonist.

The capacity of the heterodimer to interact with CoAs is also a determinant factor which can be modulated by ligand binding for activating heterodimers. By releasing CoR from RARα, CD2665 permits agonist-bound RXR to recruit CoA. However, while Ro41-5253 can destabilize CoR interaction with the heterodimer, no synergy is observed when this RAR ligand is combined to a rexinoid agonist. Our data show that Ro41-5253 totally hinders CoA interaction with RARα alone or RXR-RARα heterodimer. Furthermore, the Ro41-5253-RARα complex is less competent than unliganded RARα for CoA association. These observations suggest that, to efficiently promote CoA recruitment by RXR-RARα, liganded RXR needs RARα LBD to exhibit at least an interface on its surface for a weak interaction with a CoA LxxLL motif, as in the case of the unliganded- or CD2665 bound-RARα LBD. In such configurations, two LxxLL motifs presumably interact in a cooperative manner to each dimeric subunit. Ro41-5253 prevents this cooperativity, in agreement with the weak transcriptional activity of RXR-RARα obtained under a Ro41-5253-RXR agonist co-treatment [81], and the inability of this ligand combination to inhibit MCF-7 cell growth. Our data define Ro41-5253 as one of the most efficient retinoids preventing RXR-RARα activation.

Our results corroborate the ligand unresponsiveness of RXR, unless RAR is bound to a ligand that destabilizes the interaction with co-repressors. In this respect, the phantom ligand effect assigned to the RXR antagonist LG754 was intriguing. This postulation states that LG754 binds to RXR and functions as an allosteric regulator converting the unliganded RAR into an active receptor with high affinity for CoAs [50,51]. Rather, the LG754 activity profiles determined here show that LG754 mostly acts through direct binding to RAR. A first indication of a possible binding of LG754 to RAR in a RXR-RAR heterodimer was brought by electrospray ionization mass spectrometry analyses with a purified RXR-RAR LBD heterodimer that showed the presence of two LG754 molecules per heterodimer [69]. Together with our proteolysis and transcriptional assays, the resolution of the crystal structure of the LG754-RARβ LBD complex bound to an LxxLL peptide in the agonist conformation definitively proves the binding of LG754 to RAR and that this binding can induce an active conformation of the LBD. By doing so, LG754 allows for an efficient CoA recruitment by RARs as well as a moderate CoR release. As a result, this compound shifts the equilibrium of co-regulator interaction with the heterodimer in favor of CoA binding, accounting for its partial agonism. The partial character of LG754 makes it possibly sensitive to the cellular environment, as we could observe in different cel- based models (data not shown). The LG754-induced displacement of the co-regulator interaction balance is even more pronounced when LG754 is combined with a pure RXR agonist, which, at high concentration, can displace LG754 from RXR, and thus synergize with LG754 on the RAR side. It should be noted that LG754 resembles LE135 in terms of RARγ functional profile. On the other hand, our comparative studies of LG754 and the pure rexinoid UVI3002 suggest that these molecules are structurally and functionally similar. This implies that, if an allosteric effect was driven by LG754, UVI3002 should exert a similar one. However, UVI3002 is inactive on RXR-RAR, except those associated with an RAR agonist with whom it cooperates up to its RXR partial agonistic level. The only significant difference between both compounds was revealed by our structural data demonstrating that, in contrast to UVI3002 and like 9CRA [71], LG754 can adapt to both RXR and RAR LBDs. Overall, our data defines LG754 as a dual ligand exerting a strong RAR and a weak RXR agonistic activity. As such, LG754 behaves as a potent RXR-RAR heterodimer activator.

Altogether, the above considerations related to our in-depth retinoid characterization suggest several caveats regarding the experimental use of these retinoids. This issue is of prime importance as it concerns commercially available and widely used retinoids. Beyond the attention that must be paid to the affinity of a ligand towards the different receptors to achieve a desired specificity, it is also necessary to consider the precise effect of a ligand on the interaction of co-regulators with the receptor. For instance, BMS493 should not be used as a mere antagonist. Contrary to neutral antagonists such as BMS614 [21,22], BMS493 may affect signaling pathways by inducing a strong transcriptional repression of retinoid-target genes, and this could be dramatic in the context of both cellular and animal experiments. It is also necessary to apprehend the ability of an RAR antagonist to transcriptionally synergize with an RXR agonist, as illustrated by CD2665 and AGN870. In addition, CD2665 is described as an RARγ/β selective antagonist while our data reveal that it also acts as an RARα antagonist, and, above all, it can cooperate with an RXR agonist to activate RARα-mediated pathways. Our work provides also an in-depth characterization of LE135. While this molecule is reported as a selective antagonist of RARβ [82], we clearly show that LE135 can act as an RARγ partial agonist and promote CoR release from this receptor. This surprising discrepancy in LE135 definition can be due to differences in experimental conditions, but the morphological differentiation of F9 cells induced by LE135 confirms its RARγ profile. Other parameters should be also considered as the ability of a retinoid to bind another nuclear receptor. This issue is highlighted by the case of the RARα antagonist Ro41-5253 that has been demonstrated to activate PPARγ [66]. Our demonstration of the RAR binding capacity of LG754 is another dramatic case of such a possibility. Taken together, these examples stress the importance to take into account all properties of a retinoid in order to avoid misinterpretation of experiments. In addition, previous results may be reconsidered in view of the novel retinoid functionalities reported in the present study.

## Figures and Tables

**Figure 1 cells-08-01392-f001:**
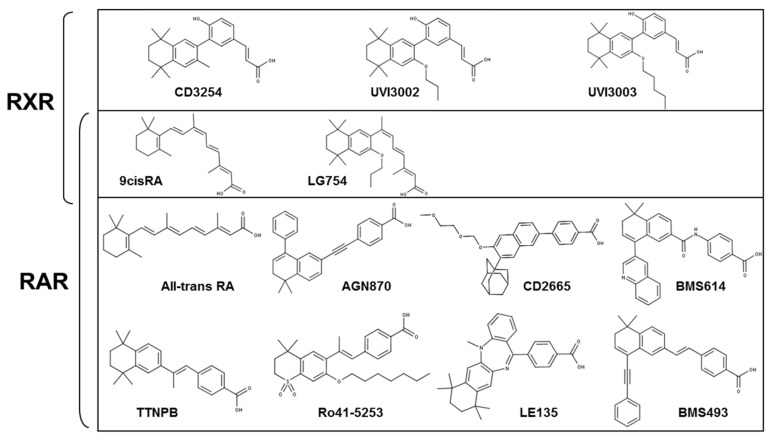
Chemical structures of the retinoids and rexinoids used in this study.

**Figure 2 cells-08-01392-f002:**
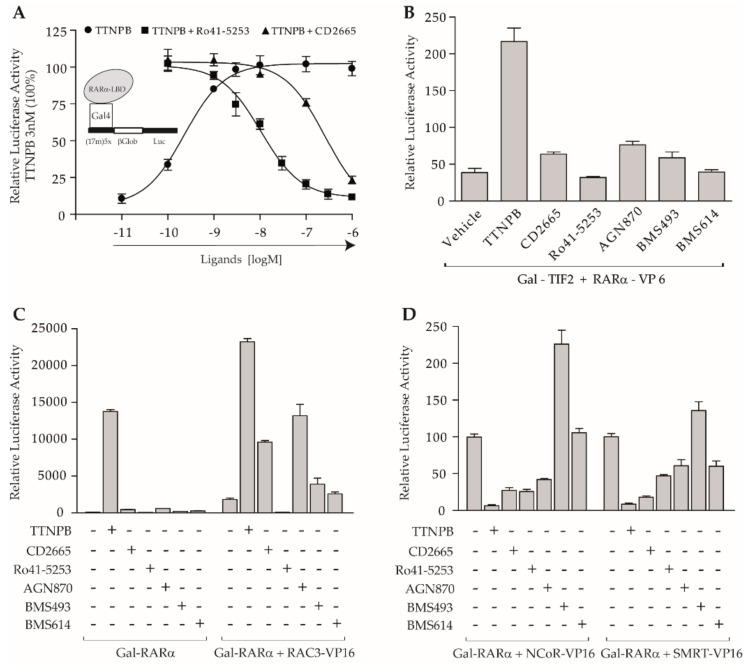
Co-regulator interactions define different classes of RARα antagonists. Transient transfections were performed in COS cells; (**A**) to assess antagonistic activities of CD2665 and Ro41-5253 cells were co-transfected with (17m)5x-βGlob-Luc and Gal-RARα. The reporter was activated by 3 nM TTNPB (100%) and increasing concentrations of CD2665 (closed triangles) or Ro41-5253 (closed squares) were added in a concentration range of 10^−10^ to 10^−6^ M, as indicated. To assess the agonistic activity of TTNPB, cells were incubated with increasing concentrations of TTNPB alone in a range of 10^−11^ to 10^−6^ M (closed circles); (**B**) mammalian two-hybrid assay with (17m)5x-βGlob-Luc and Gal-TIF2 as bait and VP16-RARα as prey was performed to assess the influence of indicated retinoids on interaction between RARα and TIF2; (**C**) mammalian two-hybrid assays with Gal-RARα and RAC3-VP16 (right panel) were performed to reveal the partial agonist activity of RARα antagonists. Gal-RARα used alone (left panel); (**D**) mammalian two-hybrid assays with (17m)5x-βGlob-Luc and VP16-RARα and Gal-NCoR (left panel) or Gal-SMRT (right panel) were performed to assess the influence of indicated retinoids on interaction between RARα and CoRs (100%, basal transcriptional activity). Compounds were used at 1 µM in all two-hybrid assays. Error bars, s.e.m.

**Figure 3 cells-08-01392-f003:**
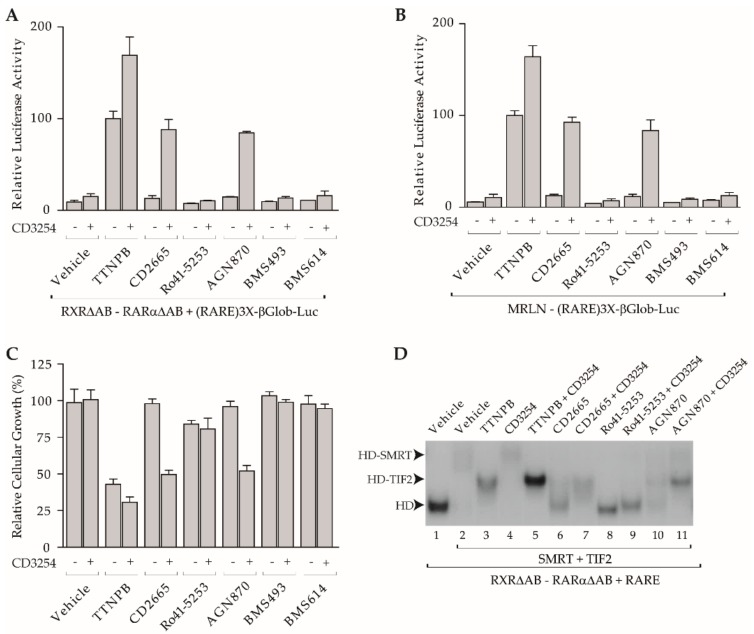
CD2665 and AGN870 synergize with CD3254 to activate RXR-RARα-mediated pathways. In the following cellular assays, compounds were used at 1 µM; (**A**) transient transactivation in COS cells co-transfected with the reporter (RARE)3x-tk-Luc and RARαΔAB and RXRΔAB expression vectors in the absence or in the presence of indicated retinoids used alone or combined with CD3254 reveals synergy between CD3254 and CD2665 or AGN870 (100%, TTNPB induced activity). Error bars, s.e.m.; (**B**) transactivation assays in MRLN cells stably transfected with the reporter (RARE)3x-tk-Luc in the absence or in the presence of indicated retinoids used alone or with CD3254 reveals synergy between CD3254 and CD2665 or AGN870 through endogenous RXR-RARα (100%, TTNPB induced activity). Error bars, s.e.m.; (**C**) effects of indicated retinoids used alone or with CD3254 on anchorage-dependent growth of MCF-7 mammary carcinoma cells in culture after six days compared to the non-treated control (100%).Error bars, s.e.m., (**D**) EMSAs demonstrating ligand-dependent co-regulator recruitment by the RARαΔAB- RXRΔAB heterodimer (HD). RARE oligonucleotide, HD, SMRT, and TIF2 were co-incubated in the absence or in the presence of saturated amounts of indicated retinoids used alone or with CD3254.

**Figure 4 cells-08-01392-f004:**
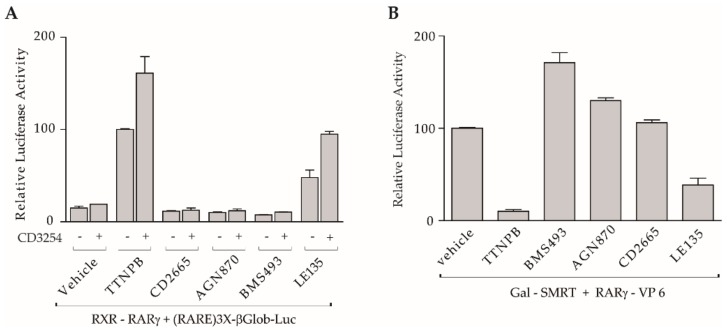
CD2665 and AGN870 do not synergize with CD3254 to activate RXR-RARγ. (**A**) transient transactivation in COS cells co-transfected with the reporter (RARE)3x-tk-Luc and RARγ and RXR expression vectors in the absence or in the presence of indicated retinoids used alone or with CD3254 (100%, TTNPB induced activity); (**B**) mammalian two-hybrid assay with (17m)5x-βGlob-Luc and VP16-RARγ and Gal-SMRT was performed in COS cells to assess the influence of indicated retinoids on interaction between RARγ and CoRs (100%, basal transcriptional activity). Compounds were used at 1 µM in all two-hybrid assays. Error bars, s.e.m.

**Figure 5 cells-08-01392-f005:**
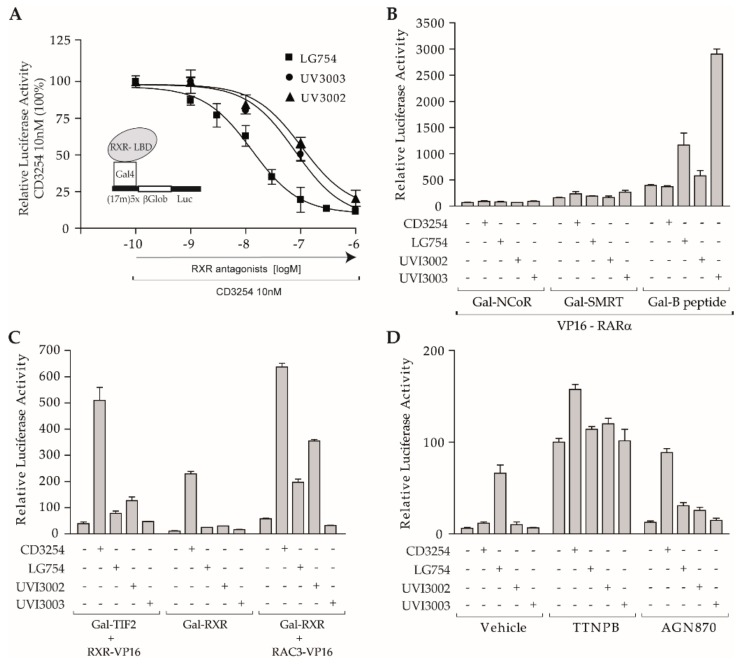
Contribution of different classes of rexinoids in the RXR-RARα heterodimer activity. (**A**) To assess antagonistic activities of synthetic rexinoids, COS cells were co-transfected with (17m)5x-βGlob-Luc and Gal-RXR expression vector. The reporter was activated by 10 nM CD3254 (100%) and increasing concentrations of LG754 (closed squares) or UVI3002 (closed triangles) or UVI3003 (closed circles) were added in a concentration range of 10^−10^ to 10^−6^ M, as indicated; (**B**) mammalian two-hybrid assays with (17m)5x-βGlob-Luc and VP16-RXR and Gal-NCoR (left panel) or Gal-SMRT (middle panel) or Gal-B peptide were performed in COS cells to assess the influence of indicated retinoids on interaction between RXR and CoRs. Compounds were used at 1 µM; (**C**) mammalian two-hybrid assays with (17m)5x-βGlob-Luc and Gal-TIF2 and VP16-RXR (left panel) or with Gal-RXR and RAC3-VP16 (right panel) were performed in COS cells to assess the influence of indicated retinoids on interaction between RXR and TIF2 and to reveal the partial agonist activity of RXR antagonists. Transactivation assay with Gal-RXR used alone is shown in the middle panel. Compounds were used at 1 µM; (**D**) transactivation assays in MRLN cells to assess the activity of the indicated rexinoids on RXR-RARα activity when used alone or in association with either TTNPB or AGN870 (100%, TTNPB induced activity). Compounds were used at 1 µM. Error bars, s.e.m.

**Figure 6 cells-08-01392-f006:**
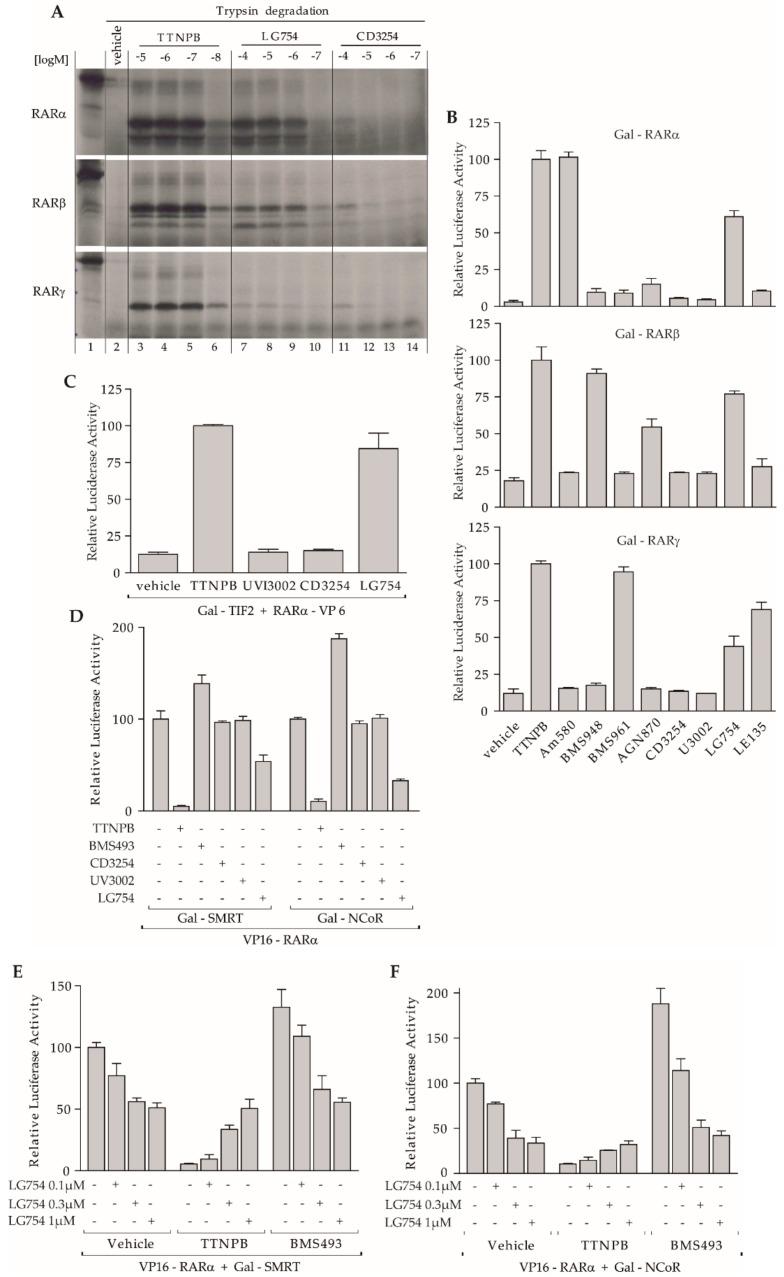
LG754 is a partial agonist for all three RARs. (**A**) RAR protease sensitivity reveals LG754 binding to RARs. Partial proteolysis maps of in vitro-translated RARs in the presence or the absence either of TTNPB, LG754, or CD3254 at the indicated concentrations. Radiolabeled RARs were exposed to trypsin for 10 min. (Top) RARα, (Middle) RARβ, (Bottom) RARγ; (**B**) COS cells were transiently co-transfected with the reporter (17m)5x-βGlob-Luc and Gal-RARα (Top), Gal-RARβ (Middle) or Gal-RARγ (Bottom), as indicated, to assess the RAR agonist potential of LG754. Cells were incubated with specific agonists for each RAR at selective concentrations (Am580 (RARα) 10^−9^ M, BMS948 (RARβ) 10^−6^M, BMS961 (RARγ) 10^−7^ M). Other compounds were used at 1 µM; 100% corresponds to the reporter gene transcription induced in the presence of the full pan-RAR agonist TTNPB; (**C**) mammalian two-hybrid assay with (17m)5x-βGlob-Luc and Gal-TIF2 and VP16-RARα in COS cells to assess the influence of LG754 on interaction between RARα and TIF2 (100%, TTNPB); (**D**) mammalian two-hybrid assays with (17m)5x-βGlob-Luc and VP16-RARα and Gal-SMRT (left panel) or Gal-NCoR (right panel) in COS cells to assess the influence of LG754 on interaction between RARα and CoRs (100%, basal transcriptional activity); (**E**,**F**) Competitive activity of increasing concentrations of LG754 on RAR interaction with CoR either induced by TTNPB or BMS493 at 10 nM. SMRT (**E**) or NCoR (**F**).

**Figure 7 cells-08-01392-f007:**
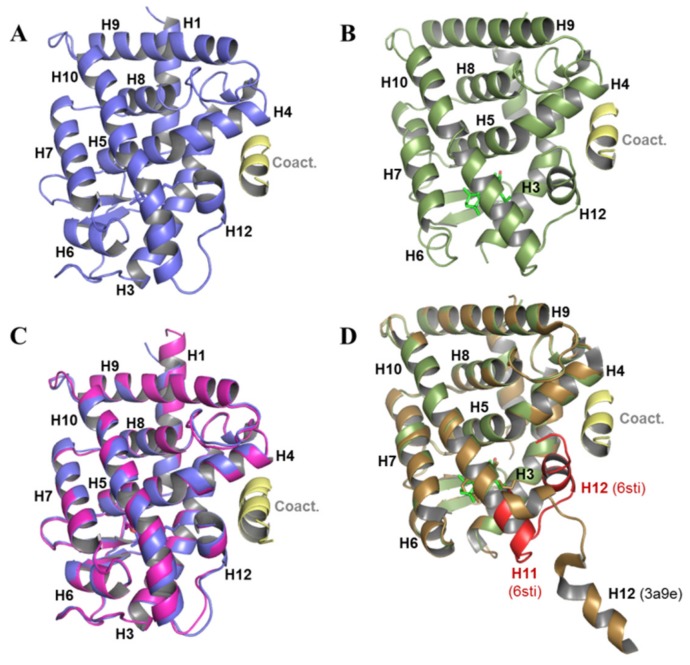
3D crystal structure of RARβ and RXRα LBDs in complex with LG754 and a coactivator peptides. (**A**) cartoon representation of RARβ LBD. Helices are numbered from H1 to H12, the latest being the activation helix that adopts the agonist conformation. The coactivator peptide (SCR1 NR2) bound to the surface formed by H3, H4, and H12 of RARβ is shown in yellow. The ligand LG754 is shown by stick representation; (**B**) cartoon representation of RXRα LBD. The coactivator peptide (TIF2 NR2) bound to the protein is shown in yellow; (**C**) superposition of the two complexes of RARβ LBD bound to LG754 and coactivator peptide observed in the asymmetric unit; (**D**) superposition of our complex of RXRα LBD bound to LG754 and coactivator peptide (same as in B) with the crystal structure of RXRα LBD bound to LG754 in the context of the heterodimer RARα-RXRα [69] (PDB code 3a9e). In the latest structure, the activation helix H12 adopts an antagonist position protruding outside the LBD, whereas, in the present structure (PDB code 6sti), helix H12 (shown in red) folds back on the LBD in the agonist position.

**Figure 8 cells-08-01392-f008:**
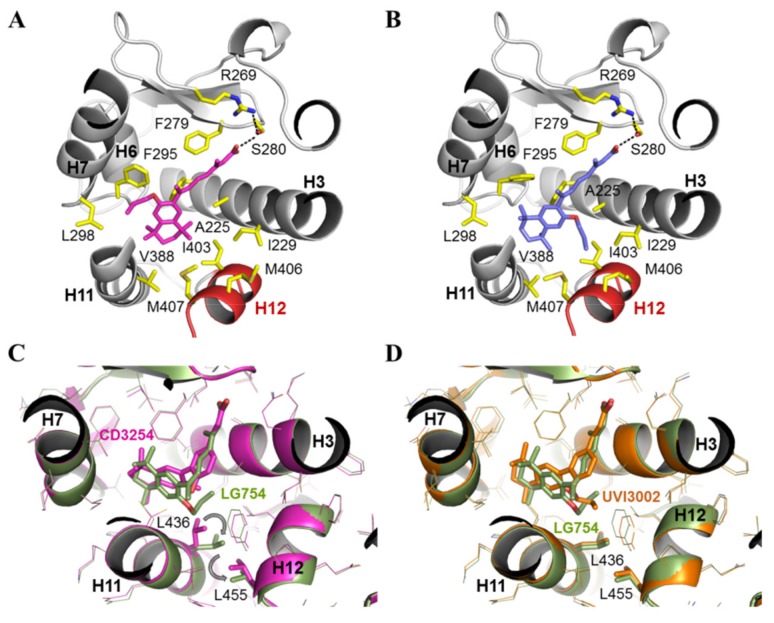
Structural basis of LG754 activity in RARβ and RXRα. (**A**,**B**) close-up view of LG754 in the LBP of RARβ as observed in the two molecules of the asymmetric unit. Side chains of residues in close contact with the ligand are shown as yellow sticks. In both cases, hydrogen bonds between the carboxylate group of LG754, R269 in H5 and S280 in the β-turn are highlighted in dashed black line. The propoxy group of the ligand points either toward helices H6 and H7 (in a) or toward helix H12 (in b); (**C**) close-up view of the superposition of RXRα LBP bound to LG754 (in green, PDB code 6sti) with RXRα LBP bound to CD3254 (in pink, PDB code 3fug) [68]. The propoxy group of LG754 pointing toward H12 induces a reorientation of L436 from helix H11, as compared to its position in the RXRα-CD3254 complex, generating a steric clash with L455 of helix H12; (**D**) close-up view of the superposition of RXRα LBP bound to LG754 (in green, PDB code 6sti) with RXRα LBP bound to UVI3002 (in orange, PDB code 2p1v) [54]. The aliphatic extension of UVI3002 induces the same positioning of L436 and L455, as observed in the structure of RXRα bound to LG754.

**Figure 9 cells-08-01392-f009:**
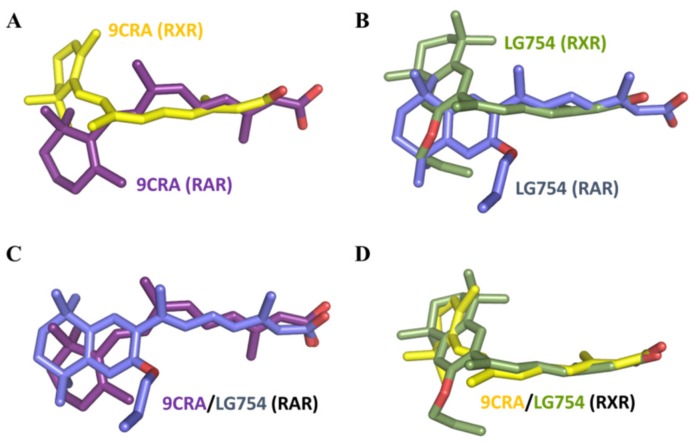
Conformation of the bound LG754 in RARβ and RXRα. (**A**) superposition of 9CRA as seen in the LBP of RXRα (PDB code 1fby) [70] and RARβ (PDB code 1xdk) [71]; (**B**) superposition of LG754 as seen in the LBP of RXRα (PDB code 6sti) and RARβ (PDB code 6ssq); (**C**) superposition of LG754 and 9CRA as seen in the LBP of RARβ (PDB code 6ssq) and of 9CRA as seen in the LBP of RARβ (PDB code 1xdk) [71]; (**D**) superposition of LG754 as seen in the LBP of RXRα (PDB code 6sti) and of 9CRA as seen in the LBP of RXRα (PDB code 1fby) [70].

**Table 1 cells-08-01392-t001:** Synergistic effect of synthetic retinoids on F9 cell differentiation.

Retinoid Concentrations	Morphological Differentiation (72 h) ^a^
Ethanol (vehicle)	None
TTNPB (10 nM)	+++
CD3254 (0.1 µM)	None
CD3254 (3 µM)	None
Am580 (1 nM)	None
BMS961 (0.1 µM)	+++
LG754 (0.5 µM)	+
LG754 (0.5 µM) and CD3254 (3 µM)	+++
CD2665 (1 µM)	None
CD2665 (1 µM) and CD3254 (0.1 µM)	None
LE135 (2 µM)	+
LE135 (2µM) and CD3254 (0.1 µM)	++
AGN870 (0.1 µM)	None
AGN870 (0.1 µM) and CD3254 (0.1 µM)	none

**^a^** The morphological differentiation of the cells was monitored for 72 h after retinoid treatment. +++, ++, and + indicate that 70 to 90, 40 to 70, and < 10% of the cell population appeared differentiated at 96 h, respectively.

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
