# Peer review of "Regulation of RXR-RAR Heterodimers by RXR- and RAR-Specific Ligands and Their Combinations"

_cells, 2019, doi:10.3390/cells8111392_

Round 1

Reviewer 1 Report

The manuscript titled “Regulation of RXR-RAR heterodimers by RXR-and RAR-specific ligands and their combinations” by le Maire et al is a nice blend of basic and translational science. It addresses the functional activity of different pharmacological classes of RAR- and RXR- specific ligands alone or in combinations using biochemical, cellular and crystallographic approaches. From the results obtained, the authors propose novel molecular mechanisms at the basis of ligand- and co-regulator binding specificity and activity of RXR-RAR heterodimers. They further highlight novel functional features allowing the fine tuning of RXR- RAR heterodimer by previously studied retinoids and rexinoids. 

Overall much of the authors’ key points are novel, logic and well supported by the experimental data.

Specific comments:

The response to RAR- and RXR- specific ligands alone or in combinations may vary in different cell types. This point should be further considered and discussed. For example, I am not sure that the results obtained in COS cells (Figure 4) can be transferred to F9 cell (Table 1), where only the differentiation response to these agents was tested. 

In paragraph 3.6, page 12, line 475 do the authors mean Fig. 5D rather than 5C?

Author Response

The manuscript titled “Regulation of RXR-RAR heterodimers by RXR-and RAR-specific ligands and their combinations” by le Maire et al is a nice blend of basic and translational science. It addresses the functional activity of different pharmacological classes of RAR- and RXR- specific ligands alone or in combinations using biochemical, cellular and crystallographic approaches. From the results obtained, the authors propose novel molecular mechanisms at the basis of ligand- and co-regulator binding specificity and activity of RXR-RAR heterodimers. They further highlight novel functional features allowing the fine tuning of RXR- RAR heterodimer by previously studied retinoids and rexinoids. 

Overall much of the authors’ key points are novel, logic and well supported by the experimental data.

Specific comments:

The response to RAR- and RXR- specific ligands alone or in combinations may vary in different cell types. This point should be further considered and discussed. For example, I am not sure that the results obtained in COS cells (Figure 4) can be transferred to F9 cell (Table 1), where only the differentiation response to these agents was tested. 

As specified in the text (page 10, below Figure 4), it has been previously shown that the induction of F9 differentiation into primitive endoderm is closely correlated to the activation of the RXR-RARγ heterodimer (reference 67). On the other hand, we have established a F9-derived cell line, similar to MRLN cell line in the context of MCF-7 cells, which contains a stably transfected luciferase reporter system (RARE)3x-tk-Luc (data not shown). A pattern of ligand activation similar to that observed with COS cells transiently transfected with RARγ (Figure 4A) was obtained with the FRLN cells, namely for example that CD2665 is unable to transcriptionally synergize with an RXR agonist, which is in line with the inability of this ligand combination to elicit F9 differentiation.

In paragraph 3.6, page 12, line 475 do the authors mean Fig. 5D rather than 5C?

Mention to Figure 5C has been removed. Figure 5D is cited at the end of the sentence (page 13).

Reviewer 2 Report

General comment:

The author provide a thorough and detailed analysis of RAR-RXR heterodimer transcriptional regulation using a set of synthetic ligands. With this work, the authors reassessed, confirmed and/or put forward novel regulatory mechanisms. The experiments are sound, well-planned and rigorous and the presented dataset provides important clues for the study of RAR-RXR signalling.

Specific comments:

I find the introduction is a bit too long. If possible move some of the more specific detail to the discussion: i.e. the description of CoA and CoR motifs.

Regarding Figure 2: while I understand the use of the combination GAL-TIF2/RARα-VP16 (which should be further clarified in the text, see below), in order to address the interaction between RARα/TIF2 (2B), I wonder if similar trends and activation levels would have been observed using the other combination, that is GAL-RARα/TIF2-VP16, as shown for RAC3 (or in Figure 5C for RXR). In fact, I am intrigued by the activation levels obtained for GAL-TIF2/RARα-VP16, notably with TTNPB. When GAL-RARα/RAC3-VP16 was exposed to TTNPB, luciferase activity was incremented, when compared to transfection with GAL-RARα alone, which promotes ligand-dependent transcription, as expected (2C). Yet, despite being an independent assay, the activation obtained for GAL-TIF2/RARα-VP16 + TTNPB seems quite low when compared to GAL-RARα alone.

Lines 48-51: sentence too long, maybe split in two? “(…) in the presence of some antagonists [20,21]. However, when released from CoRs (…)”

The high number of acronyms make the text sometimes hard to read. Please make sure the acronym usage is uniform and clear: i.e. line 72 “CoRNR box1” and in line 75 “CoRNR1”

Line 60: “TIF2, SRC2 or GRIP1”, it should be clear these are aliases

Line 61: “RAC3, SRC3 or AIB1”

Line 123: in the “Materials and chemicals” section it should be clear which vectors include partial or full sequences: i.e. Gal- RARαLBD instead of Gal- RARα (as specified in the subsequent sections: i.e line 228, VP16-RARαLBD). Make sure the abbreviations are consistent throughout the text.

Line 223: “allowed the definition”

Line 228. To make the article more amenable to a broader audience, it would probably be useful to add a line to clarify the GAL4/VP16 system, specifying that, GAL4-TIF2 alone, albeit able to bind the corresponding DNA response element, is unable to yield ligand-dependent luciferase activity.

Line 321-322:  “(..) proliferation, which is in line (…)

Line 337: “bound to the”

Line 345-348: “Interestingly, despite decreasing the amount of bound SMRT, Ro41-5253 completely blocked…”

Line 433: “CoR binding, meaning” (comma)

Line 440: “relies”

Line 483: “As a result,” (comma)

Line 669: “data confirms”

Line 681: “allowed us to demonstrate”

Line 684: “binding” remove s

Line 687: “In this respect,” (comma)

Line 690: “which depend” remove s

Line 714: “However,” (comma)

Line 721: “complex”

Line 737: “Accordingly,” (comma)

Line 742: “Other retinoids” remove s

Line 751: “interaction with the heterodimer”

Line 753: replace Even

Line 774: “shifts”

Line 776: “makes it possibly sensitive to”

Line 787: “defines”

Author Response

General comment:

The author provide a thorough and detailed analysis of RAR-RXR heterodimer transcriptional regulation using a set of synthetic ligands. With this work, the authors reassessed, confirmed and/or put forward novel regulatory mechanisms. The experiments are sound, well-planned and rigorous and the presented dataset provides important clues for the study of RAR-RXR signalling.

Specific comments:

I find the introduction is a bit too long. If possible move some of the more specific detail to the discussion: i.e. the description of CoA and CoR motifs.

We agree with the reviewer that the introduction is quite prolix but we feel that it provides the necessary background so the article can be accessible to a large non-specialized readership. For instance, knowledge about CoA and CoR motifs is necessary to understand how experiments are designed and interpreted.      

Regarding Figure 2: while I understand the use of the combination GAL-TIF2/RARα-VP16 (which should be further clarified in the text, see below), in order to address the interaction between RARα/TIF2 (2B), I wonder if similar trends and activation levels would have been observed using the other combination, that is GAL-RARα/TIF2-VP16, as shown for RAC3 (or in Figure 5C for RXR). In fact, I am intrigued by the activation levels obtained for GAL-TIF2/RARα-VP16, notably with TTNPB. When GAL-RARα/RAC3-VP16 was exposed to TTNPB, luciferase activity was incremented, when compared to transfection with GAL-RARα alone, which promotes ligand-dependent transcription, as expected (2C). Yet, despite being an independent assay, the activation obtained for GAL-TIF2/RARα-VP16 + TTNPB seems quite low when compared to GAL-RARα alone.

Indeed, the two systems used are different and for unknown reasons, the levels of activation (and of induction) measured with the GAL-RARα/RAC3-VP16 is always superior. It should be noted that the two-hybrid system using the fusion proteins RARα LBD-VP16 and Gal-TIF2 NRID never produces high induction in our tests, even in the presence of an agonist like TTNPB. Importantly, the general trend remains whatever the system used and when comparing ligand activities within each system. It can be anticipated that a same activation profile would be observed by replacing RAC3 NRID-VP16 with TIF2 NRID-VP16.

Lines 48-51: sentence too long, maybe split in two? “(…) in the presence of some antagonists [20,21]. However, when released from CoRs (…)”

The sentence has been split into two (page 2).

The high number of acronyms make the text sometimes hard to read. Please make sure the acronym usage is uniform and clear: i.e. line 72 “CoRNR box1” and in line 75 “CoRNR1”

CoRNR1,2,3 are now used throughout the manuscript.

Line 60: “TIF2, SRC2 or GRIP1”, it should be clear these are aliases

Line 61: “RAC3, SRC3 or AIB1”

Change has been made (page 2)

Line 123: in the “Materials and chemicals” section it should be clear which vectors include partial or full sequences: i.e. Gal- RARαLBD instead of Gal- RARα (as specified in the subsequent sections: i.e line 228, VP16-RARαLBD). Make sure the abbreviations are consistent throughout the text.

The domains contained in the constructs have been specified as suggested (page 3).

Line 223: “allowed the definition”

Change has been made (page 5).

Line 228. To make the article more amenable to a broader audience, it would probably be useful to add a line to clarify the GAL4/VP16 system, specifying that, GAL4-TIF2 alone, albeit able to bind the corresponding DNA response element, is unable to yield ligand-dependent luciferase activity.

A few lines explaining how the two-hybrid assay actually works has been added (page 5).

Line 321-322:  “(..) proliferation, which is in line (…)

Correction has been made (page 9).

Line 337: “bound to the”

Correction has been made (page 9).

Line 345-348: “Interestingly, despite decreasing the amount of bound SMRT, Ro41-5253 completely blocked…”

Correction has been made (page 9).

Line 433: “CoR binding, meaning” (comma)

Correction has been made (page 11).

Line 440: “relies”

Correction has been made (page 12).

Line 483: “As a result,” (comma)

Correction has been made (page 13).

Line 669: “data confirms”

Correction has been made (page 18).

Line 681: “allowed us to demonstrate”

Correction has been made (page 18).

Line 684: “binding” remove s

Correction has been made (page 18).

Line 687: “In this respect,” (comma)

Correction has been made (page 18).

Line 690: “which depend” remove s

Correction has been made (page 18).

Line 714: “However,” (comma)

Correction has been made (page 19).

Line 721: “complex”

Correction has been made (page 19).

Line 737: “Accordingly,” (comma)

Correction has been made (page 19).

Line 742: “Other retinoids” remove s

Correction has been made (page 19).

Line 751: “interaction with the heterodimer”

Correction has been made (page 19).

Line 753: replace Even

We have replaced “Even” by “Furthermore” (page 19).

Line 774: “shifts”

Correction has been made (page 20).

Line 776: “makes it possibly sensitive to”

Correction has been made (page 20).

Line 787: “defines”

Correction has been made (page 20).

Reviewer 3 Report

This paper describes a series of experiment in which the regulation of RXR-RAR heterodimers is probed for ligand-induced interactions.  The interactions are demonstrated to regulate the interaction and/or recruitment of coA and CoR molecules that in vivo can determine the precise transcriptional regulation.  Identifiying these interactions allowed the researchers to determine which "antagonists" fail to release CoR, and which fail to recruit CoA.  Analyses demonstrate that both the ligand bound to the RXR and the RAR proteins determine the exact TF components on the promoter region that in turn "fine tune" the transcriptional levels.  This work is crucial for our continued understanding of HOW exactly RXR binds to its heteropartners and elicits differential transcriptional effects.  This work sheds light on our understanding of when and how each heteropartner yeilds to the ligand and binding of the others.   

Figure 7 was difficult to analyze as the coactivator peptide and the RAR beta LBD and the CoA peptides are in the same color family.  I would suggest that the LBD and the CoA peptides in this figure are in contrasting colors.

Several places in figure lgends there is the copyright symbol (c) where the authors meant to place a C).  

Author Response

This paper describes a series of experiment in which the regulation of RXR-RAR heterodimers is probed for ligand-induced interactions.  The interactions are demonstrated to regulate the interaction and/or recruitment of coA and CoR molecules that in vivo can determine the precise transcriptional regulation.  Identifiying these interactions allowed the researchers to determine which "antagonists" fail to release CoR, and which fail to recruit CoA.  Analyses demonstrate that both the ligand bound to the RXR and the RAR proteins determine the exact TF components on the promoter region that in turn "fine tune" the transcriptional levels.  This work is crucial for our continued understanding of HOW exactly RXR binds to its heteropartners and elicits differential transcriptional effects.  This work sheds light on our understanding of when and how each heteropartner yeilds to the ligand and binding of the others.

Figure 7 was difficult to analyze as the coactivator peptide and the RAR beta LBD and the CoA peptides are in the same color family.  I would suggest that the LBD and the CoA peptides in this figure are in contrasting colors.

The CoA peptides are now shown in yellow in all figures.

Several places in figure lgends there is the copyright symbol (c) where the authors meant to place a C).

This has been corrected in the legends of Figures 8 and 9.